# The Effect of Tortuosity on Permeability of Porous Scaffold

**DOI:** 10.3390/biomedicines11020427

**Published:** 2023-02-01

**Authors:** Akbar Teguh Prakoso, Hasan Basri, Dendy Adanta, Irsyadi Yani, Muhammad Imam Ammarullah, Imam Akbar, Farah Amira Ghazali, Ardiyansyah Syahrom, Tunku Kamarul

**Affiliations:** 1Doctoral Program Study of Engineering Science, Faculty of Engineering, Universitas Sriwijaya, Palembang 30139, South Sumatra, Indonesia; 2Department of Mechanical Engineering, Faculty of Engineering, Universitas Sriwijaya, Indralaya 30662, South Sumatra, Indonesia; 3Department of Mechanical Engineering, Faculty of Engineering, Pasundan University, Bandung 40153, West Java, Indonesia; 4Biomechanics and Biomedics Engineering Research Centre, Pasundan University, Bandung 40153, West Java, Indonesia; 5Undip Biomechanics Engineering & Research Centre (UBM-ERC), Diponegoro University, Semarang 50275, Central Java, Indonesia; 6Department of Mechanical Engineering, Faculty of Engineering, Tridinanti University, Palembang 30129, South Sumatra, Indonesia; 7Applied Mechanics and Design, Faculty of Mechanical Engineering, Universiti Teknologi Malaysia, Skudai 81310, Johor Bahru, Malaysia; 8Medical Device and Technology Center (MEDiTEC), Institute of Human-Centered and Engineering (IHumEn), Universiti Teknologi Malaysia, Skudai 81310, Johor Bahru, Malaysia; 9Tissue Engineering Group, National Orthopaedic Centre of Excellence for Research and Learning (NOCERAL), Faculty of Medicine, University of Malaya, Kuala Lumpur 50603, Malaysia; 10Advanced Medical and Dental Institute (AMDI), Universiti Sains Malaysia, Bertam 13200, Kepala Batas Pinang, Malaysia

**Keywords:** bone tissue engineering, porous scaffold, tortuosity, permeability, computational fluid dynamics

## Abstract

In designing porous scaffolds, permeability is essential to consider as a function of cell migration and bone tissue regeneration. Good permeability has been achieved by mimicking the complexity of natural cancellous bone. In this study, a porous scaffold was developed according to the morphological indices of cancellous bone (porosity, specific surface area, thickness, and tortuosity). The computational fluid dynamics method analyzes the fluid flow through the scaffold. The permeability values of natural cancellous bone and three types of scaffolds (cubic, octahedron pillar, and Schoen’s gyroid) were compared. The results showed that the permeability of the Negative Schwarz Primitive (NSP) scaffold model was similar to that of natural cancellous bone, which was in the range of 2.0 × 10^−11^ m^2^ to 4.0 × 10^−10^ m^2^. In addition, it was observed that the tortuosity parameter significantly affected the scaffold’s permeability and shear stress values. The tortuosity value of the NSP scaffold was in the range of 1.5–2.8. Therefore, tortuosity can be manipulated by changing the curvature of the surface scaffold radius to obtain a superior bone tissue engineering construction supporting cell migration and tissue regeneration. This parameter should be considered when making new scaffolds, such as our NSP. Such efforts will produce a scaffold architecturally and functionally close to the natural cancellous bone, as demonstrated in this study.

## 1. Introduction

Tissue engineering (TE) scaffolds are functional replacements for bone defects that use biological and engineering principles [1,2,3,4,5]. In orthopedic applications, TE scaffolds placed right into the injury site help the bone heal through regenerative processes. From an engineering point of view, one of the main challenges in developing TE scaffolds is optimizing the scaffold design to meet biological requirements. It includes considering the ability to support cell seeding, cell differentiation, cell proliferation, and vascularization [6,7,8,9,10]. Therefore, an essential parameter in designing a scaffold must take into account the histomorphometric characteristics of the scaffold so that the requirements mentioned above can be met. Based on biomechanical theory and mass transport phenomena, it has been found that it is essential to control several parameters, such as porosity, interconnectivity, surface curvature, tortuosity, pore size, and shape. Accordingly, it is indispensable to consider the parameters described above when designing scaffolding.

Many studies have shown that tortuous models during tissue growth promote better cell anchoring and tissue repair [11,12,13,14]. Furthermore, it has been shown that scaffolds with tortuous architecture provide better cell attachment than scaffolds with relatively straight microchannels [13]. In addition, the microstructure with tortuous surfaces gives rise to an excellent surface-to-volume ratio by significantly increasing cell surface contact. It indicates a prerequisite for the successful development of network engineering constructs [15]. The additional point is that increasing flow resistance to transport liquid molecules will provide better transit time for essential molecules such as nutrients and oxygen [11,16].

While the role of scaffold permeability has been considered necessary in bone replacement design, the tortuosity’s importance has yet to be further elaborated [17,18]. It has been shown that permeability is positively related to porosity or pore size but negatively related to an increase in the surface area of the scaffold [19,20,21]. In addition, based on the fact that although tortuosity is directly related to permeability, controlling one of the parameters using a simple approach has yet to be carried out. So far, the direct relationship between tortuosity and permeability of scaffold structures has yet to be studied before and therefore requires further investigation.

This study hypothesizes that tortuosity can be affected by changing the pore size and radius of curvature to increase porosity and perfusion pressure and reduce fluid permeability. Furthermore, the specific surface area may play a role in this. In order to prove the above hypothesis, an open porous scaffold model was developed resulting from computer simulations based on tortuous microchannels to investigate tortuous phenomena. By changing the size of the scaffold, the results of the parameters involved can be predicted.

## 2. Materials and Methods

### 2.1. Parametric Design of Tortuous Microchannel in Scaffold

A bovine lateral femoral condyle was freshly gathered from a local slaughterhouse for a cancellous bone sample. The bovine lateral femoral condyle was cut into rectangular specimens using a low-speed (150 rpm) diamond saw (Behringer GmbH, Type SLB 230 DG HA, Kirchardt, Germany) under constant lubricant irrigation to minimize structural breakage and heat generation. Saline water was used as a lubricant to ensure that the temperature did not exceed 45 °C to protect the cancellous bone sample from heat damage. The sample temperature was constantly observed based on the blade and coring bit using Fluke 62 Mini Infrared Thermometer (FLUKE Europe, Eindhoven, The Netherlands). In addition, the cutting process was stopped at several stages to ensure that the temperature did not exceed a critical level. At 150–250 rpm, a 1.5 mini-thick diamond tip coring bit was used to drill the trabecular sample into a cylindrical shape with a total length 12 mm and a diameter of 10 mm. The sample was then cleaned by soaking the cancellous bone sample in chemical detergent (Pumicized, Gent-1-Kleen, New York, NY, USA) using an ultrasonic cleaner (Cest Ultrasonic, model P1100SR, Virginia, USA) for 3 h. The procedure continues with further cleansing to remove any excess bone marrow, water, and fat using water jets and air jets until any excess bone marrow cannot be detected upon visual inspection. The cancellous bone sample was then stored in an airtight plastic bag to minimize thermal cycling and put in the freezer at a temperature of −20 °C. The sample was then scanned via high-resolution micro-computed tomography (μCT) images (Skyscan 1172^™^, Bruker micro-CT, Kontich, Belgium) with a voltage source of 100 kV, a current of 100 µA and a resolution of 17.20 µm. The tomographic reconstruction of these images using Materialize Mimics^®^ software (Materialize, Wilfried, Leuven, Belgium) gives a three-dimension (3D) image volume dataset of 712 layers of two-dimension (2D) images of the cancellous bone structure. This dataset is 12 mm in height and 10 mm in diameter. The software extracted the desired region (ROI) with 60% porosity from the sample. Finally, three different scaffold structures (cubic, octahedron pillar (PO), and Schoen’s gyroid (SG)) and cancellous bone structures with a porosity (Φ) of 60% were developed and compared.

In this study, the scaffold is an open porous model with tortuous microchannels made with SolidWorks (Dassault Systèmes SolidWorks Corp., Waltham, MA, USA) software. A schematic illustration of the scaffold design stage is shown in Figure 1a. The first step is to determine the shape of the pores. It is carried out by using the idea of a meandering pore channel with different radiuses of curvature of 0.4 mm, 0.45 mm, and 0.49 mm. A Boolean subtraction function can also be used to make a new unit cell called Negative Schwarz Primitive (NSP), which has main dimensions of 2.1 mm for width, 2.1 mm for height, and 2.1 mm for length. In the last step, the individual cells are assembled to make a model bone scaffold with tortuous microchannels. Each NSP scaffold geometry was labeled according to NSP with 60% porosity and curvature of 0.4, 0.45, and 0.49 mm. Each label is assigned as NSP60r, NSP60r1, and NSP60r2 (see Figure 1b). Next, the scaffold’s porosity is changed to the required value by modifying the pore size (X). The cubic, PO, SG, and natural cancellous bone models with 60% porosity are shown in Figure 1c. A summary of bone scaffold porosity data is shown in Table 1.

### 2.2. Morphology Analysis

The scaffold morphology, including porosity and surface area, was conducted using CAD software features, and the cancellous bone was modeled using Materialize Mimics^®^ software features. 3D CAD models were exported to stereolithography (STL) format and imported into the slice software program Chitubox (CBD-Tech, Guangdong, China). The CAD model was sliced using a resolution of 17.20 µm. The remaining 244 slices and 500 × 500 pixels images were analyzed using Fiji (Image J, NIH). The trabecular thickness (Tb.Th) and separation (Tb.Sp) were calculated using the Fiji and BoneJ plugins. The image data set was then exported to MATLAB (MathWorks Corp., Natick, MA, USA) software to calculate the diffusion tortuosity. The open solver plugin Taufactor calculates the tortuosity factor based on the finite difference method (FDM) and directly uses image voxels as discretization meshes for simulation [22,23].

### 2.3. Experimental Setup

Three specimens of the NSPr1 model with a porosity of 25, 45, 60, and 65% were fabricated from polymer resins for three-dimensional stereolithography (SLA-3DP) imprinting with a resolution of 35 µm. The permeability test apparatus and the test scheme are shown in Figure 2. A peristaltic pump with a capacity of 2.16 mL/s is connected directly to the reservoir to provide continuous fluid flow. The permeability apparatus used an 8-mm inner diameter tube to connect to other equipment or sensors. The fluid flow was adjusted to 0.670 mL/min in these experiments. The value of the volumetric flow rate according to the movement of bone marrow has a flow rate range of 0.012–1.670 mL/min [24,25,26]. The mass flow rate is measured directly from the tube using an intelligent flow sensor FL0001 (EMA Electronic Ltd., New Taipei City, Taiwan). There is a specially designed chamber to hold and clamp the specimen when it is subjected to a flowing simulated body fluid (SBF). The peripheral surface of the specimen is wrapped with polytetrafluoroethylene (PTFE) tape to prevent fluid flow along the outer surface of the specimen from entering. Differential pressure is used to measure the difference in fluid pressure, whose value is used to analyze macroscopic permeability. Darcy’s law is used to calculate the permeability of the specimen. The differential pressure is measured using a pressure transducer (EMA Electronic Ltd., New Taipei City, Taiwan) with a maximum pressure of 1 bar. The USB-6009 instrument data acquisition system (DAQ) (National Instruments Company, Austin, TX, USA) was used to read, collect and store data during the test. The pressure transducer and flow rate sensor are linked to the data acquisition system of the DAQ instrument.

### 2.4. Fluid Properties and Boundary Conditions in CFD

Computational simulations were performed using COMSOL Multiphysics^®^ software (COMSOL, Inc., Burlington, MA, USA). The fluid domain uses the Boolean subtraction method when using SolidWorks^®^ software. This domain is exported in IGES format and imported into COMSOL Multiphysics^®^ software. The boundary conditions are defined so that the volumetric flow rates on the inflow and outflow sides (zero outlet pressure) are 0.67 and 0.00 mL/min, respectively. In addition, the shape is symmetrical on the lateral side of the cube and non-slip on its interior surface [20], as shown in Figure 3a. This simulation used Simulated Body Fluid (SBF) liquid with a viscosity of 1 mPa.s and a density of 1 g/cm^3^ at body temperature (37 °C) [27]. The boundary between the fluid and solid is characterized non-slip boundary during the Computational Fluid Dynamic (CFD) study where the fluid velocity at the boundary is equal to the velocity of the solid [28]. The outlet fluid pressure is set as zero. It is assumed that the fluid flow conditions are laminar and steady through the geometry of a 3D scaffold with tetrahedral elements (see Figure 3b). Convergence studies were carried out to obtain the optimal mesh size. The total number of elements for each model varies from 2,060,893 to 8,993,558. This simulation was carried out on a personal computer (Dell Precision 3630 Tower Workstation, TX, USA) with an Intel I7-8700 processor and 80 GB of RAM. This simulation uses the solver generalized minimal residual method (GMRES) as an iterative solver. First, the average pressure drop between the inflow and outflow is determined, and then the permeability is calculated according to Darcy’s law [28,29], which is shown in Equation (1),
(1)$$K=\frac{Q\times \mu \times L}{A\times \Delta P}$$
where *K* is the permeability in (m^2^), *Q* is the flow rate in (m^3^/s), μ is the fluid’s dynamic viscosity of the fluid in (Pa.s), *L* is the specimen length (m), *A* is the flow’s cross-sectional area in (m^2^), and ΔP is the pressure drop across the structure (Pa).

The tangential drag force exerted by the fluid flowing across the surface of the scaffolds, denoted by $\tau_{\omega }$ [30], is given by,
(2)$$\tau_{\omega }=\mu \frac{\partial u}{\partial h}$$
where *u* is the fluid velocity and *h* is the height along the *x*, *y*, and *z* axes [30].

### 2.5. Statistical Analysis

The permeability of scaffolds was measured three times on four samples with porosities of 25%, 45%, 60%, and 65% to accurately assess the reproducibility and repeatability of the results and compare them with simulation results. The experimental results were expressed in the average value (k) and standard deviation (σ). The statistical analysis was conducted using MINITAB software (Minitab Inc., State College, PA, USA).

## 3. Results

### 3.1. Morphology Indices

Figure 4a shows the relationship between the average trabecular thickness (Tb.Th) and separation (Tb.Sp) and porosity for the NSP scaffold model. The results showed that with increasing porosity, Tb.Sp would increase but Tb.Th would decrease. For the same porosity, by increasing the pore size and reducing the radius of curvature, the Tb.Th and Tb.Sp values of NSPr2 are higher than those of NSPr and NSPr1. For porosity of 25% to 65%, the mean values of Tb.Sp and Tb.Th NSP scaffolds were produced in the range of 0.15–0.74 mm and 0.21–0.73 mm.

Figure 4b shows the relationship between the tortuosity and porosity of a porous scaffold. The NSP model results show that the porosity increases, while its relationship to tortuosity will be explained later. For the cubic structure, the change in tortuosity was not caused by the increase in porosity. For the NSP model, when the size of the pores got more prominent and the radius of curvature of the scaffold got smaller, the tortuosity went up, but the porosity did not change. The tortuosity values for NSPr, NSPr1 and NSPr2 were 1.8, 1.6, and 1.5, respectively. It was also noticed that the tortuosity of the PO and cancellous bone scaffolds varied in different areas of the geometric plane. Interestingly, when testing the cubic model, it was found that, with increasing porosity, there was no change in tortuosity, especially for relatively straight microchannels.

The specific surface area (BS/TV) of NSP and cubic scaffolds increased with porosity up to a maximum value, which was mostly seen at 0.45 porosity values. As shown in Figure 4c, it decreased with increasing porosity. According to the previous literature, the NSP model has the same range of BS/TV values as the natural cancellous bone. In contrast, it is the opposite for the SG, PO, and cubic scaffold models [24,31,32,33,34]. Therefore, the BS/TV value for the NSPr model is higher than for other types of scaffold models. BS/TV values for the NSPr, NSPr1, NSPr2, PO, cubic, and SG models are 3.05, 2.92, 2.86, 2.54, 2.37, and 2.01 mm^−1^, respectively.

**Figure 4 biomedicines-11-00427-f004:**
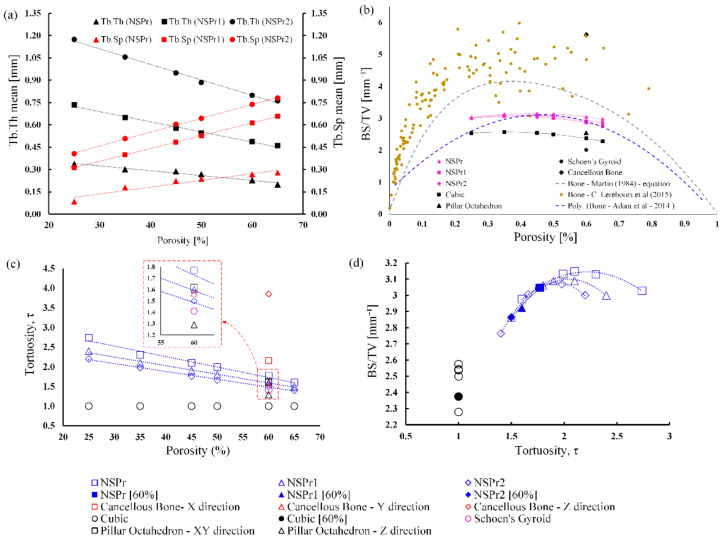
Results of morphological indices of porous scaffolds: (**a**) the relationship between BS/TV and porosity in the porous scaffold and cancellous bone (current study) and previous studies’ bone; (**b**) the relationship between porosity and tortuosity; (**c**) BS/TV versus porosity; and (**d**) the relationship between BS/TV and tortuosity [33,34].

Figure 4d shows the relationship between BS/TV and tortuosity. The correlation between BS/TV and tortuosity is nonlinear. In increasing tortuosity, while maintaining porosity, the specific surface area of the scaffold will increase. It is seen for the cubic model with 60% porosity, with the respective tortuosities of NSPr, NSPr1, NSPr2, and cubic being 1.8, 1.6, 1.5, and 1, giving specific surface areas of 3.05, 2.92, 2.86, and 2.37 mm^−1^, respectively.

### 3.2. Mesh Convergence

Convergence studies were carried out with at least 6 million fluid cells to achieve reliable results (see Figure 5). Since the permeability is directly proportional to the pressure drop, the pressure drop value from the study should be independent of the mesh density. The permeability value of the NSPr model, as determined by both experiment and simulation, is tabulated in Table 2. The permeability value determined through computer simulation was comparable to or near the value obtained from experimental measurements. The error in the permeability value between the simulation and experiment was 5%. Two essential explanations for this observation were: (1) experimental testing shows fluid flow tends to act as a turbulent medium. So, the simulation results based on Darcy’s law do not match the assumption of laminar flow; (2) it is well known that the simulated and generated geometries can be slightly different due to minor flaws in the 3D printing process. For example, the 3D-printed scaffold’s rough surface causes fluid friction and blocks fluid flow during experimental tests, which changes the pressure drop and permeability values [30].

### 3.3. Fluid Flow Characterization of Scaffold

Figure 6a shows the relationship between permeability and porosity. As the porosity increases, the permeability also increases. For each model, the porosity showed a powerful correlation to the permeability with a polynomial of 2nd order R^2^ = 0.9. Furthermore, the permeability value of cubic scaffold was higher than that of other types of scaffolds, including natural cancellous bone. For the NSP scaffold, the permeability of the NSPr2 model was higher than that of the NSPr and NSPr1 models. An increase in porosity results in an increase in permeability. The permeability values for the NSPr, NSPr1, and NSPr2 models at 60% porosity were $2.1\times 10^{-10}$ m^2^, $2.5\times 10^{-10}$ m^2^, and $3.1\times 10^{-10}$ m^2^, respectively. It indicates that the increase in porosity results in varying permeabilities. In addition, the permeability of the octahedron pillar model and natural cancellous bone is different in different planes despite having the same porosity.

Figure 6b illustrates the effect of tortuosity on permeability. It was observed that the permeability decreased with increasing tortuosity. With a polynomial of 2nd order R^2^ = 0.9, each model also found a strong correlation between tortuosity and permeability. Figure 6b also shows that each structural model, such as cubic, SG, NSP, PO, and natural cancellous bone models, has the same porosity even though the tortuosity values differ. However, it was found that there is no correlation between tortuosity and permeability for the cubic model because the structures have similar tortuosity. Therefore, it is evident that porosity is not the only parameter that contributes to permeability. For example, the magnitude of the cancellous bone tortuosity in the X, Y, and Z planes is 1.57, 2.16, and 3.85, which indicates the permeability value is $7.61\times 10^{-11}$ m^2^, $3.29\times 10^{-11}$ m^2^, and $1.06\times 10^{-11}$ m^2^, respectively.

Figure 7 shows the relationship between maximum wall shear stress (WSS) and porosity and tortuosity. WSS decreased with increasing porosity and increased with increasing tortuosity. Porosity and tortuosity strongly correlate to WSS with a polynomial of 2nd order R^2^ = 0.9 for each model tested. The WSS value for the cubic model is lower than for other scaffolds. The WSS value of the NSPr model was higher than the NSPr1, NSPr2, and cubic models. At 60% porosity, the WSS values for the NSPr, NSPr1, and NSPr2 models were 0.084 Pa, 0.069 Pa, and 0.064 Pa, respectively. It shows that the same porosity produces different WSS.

Figure 8a shows that different porous architectures with the same porosity led to different tortuosity. Cancellous bone in the Z direction showed the highest tortuosity, and cubic model scaffolds showed the lowest. The tortuosity of the cancellous bone in the Z direction is almost four times greater than that of the cubic model scaffold. Although porosity is generally considered the main factor influencing scaffold permeability [29,35,36], the findings of this study indicate that under identical porosity conditions, the tortuosity parameter is equally important (see Figure 8b). With the same structural porosity, different tortuosities are produced. The simulation results show that an increase in tortuosity and vice versa will be observed with changes in permeability. The cubic model has a permeability of 10^−9^ m in the Z direction, while the cancellous bone has a permeability of 10^−11^ m with the same porosity. The polynomial relationship between tortuosity and permeability with different architectures at the same porosity contributes R^2^ = 0.82 (see Figure 8c). The conclusion is that the change in the magnitude of the permeability of the scaffold is highly dependent on the porosity design with tortuous structures.

Figure 9 shows the velocity streamline, the velocity contour, and the pressure contour when the porosity was set to 60%. The velocity streamline of the structure illustrates the trajectory of the flow rate in the structure. The cubic scaffold streamline is straight compared with the PO and SG. The NSPr structure appeared to be the closest to natural cancellous bone. The NSPr and natural cancellous bone velocity were well distributed compared to the cubic, the pillar octahedron, and Schoen’s gyroid. The higher velocity in natural cancellous bone is seen at a particular location, namely at the bottleneck of the curvature. The lowest velocity is seen in the PO model. The maximum velocity values for SG, NSPr, cubic, PO, and natural cancellous bone were 0.59, 0.62, 0.64, 1.28, and 4.97 mm/s, respectively. The pressure pattern profile of all the scaffolds is almost the same. However, the maximum pressure value was different. The maximum pressure was in natural cancellous bone with a pressure of 10.9 Pa, and the minimum was a cubical structure with a pressure of 0.1 Pa.

Figure 10 compares the permeability values of various scaffolds with natural cancellous bone tested in the present study and the literature. Permeability values of cubic were in the range of 1.1 × 10^−10^ m^2^ to 1.5 × 10^−9^ m^2^; PO 2.9 × 10^−10^ m^2^ to 5.6 × 10^−10^ m^2^; SG 7.9 × 10^−10^ m^2^; and NSP 2.0 × 10^−11^ m^2^ to 4.0 × 10^−10^ m^2^ and cancellous bone (present study) 1.1 × 10^−11^ m^2^ to 7.6 × 10^−11^ m^2^. The permeability of cancellous bone from the literature varied from 2.5 × 10^−11^ m^2^ to 7.43 × 10^−8^ m^2^. The cancellous bone samples were taken from the vertebral body of the calcaneus [24], the femoral bone, and the spine [35,36,37]. From Figure 10, the proposed structure is within the range of natural cancellous bone.

## 4. Discussion

In bone tissue engineering (TE), the need for custom-made biomimetic scaffolds has risen sharply. It is because people want faster and better repair results all the time. It will also be essential in medical practice since the skeletal structure is defined as a person’s ability to move and support structural loads while doing daily activities. In searching for the best scaffold for use in TE, it has been found that considerable challenges in designing porous structures must be considered simultaneously, namely the need to improve geometric characteristics, mechanical properties, and permeability. Many researchers have previously claimed to have developed such a scaffold. For example, Chen et al. (2021) [39], Fantini et al. (2018) [40], and Gomez et al. (2016) [41] have built irregular scaffolds using Voronoi tessellation and shown that successful designs have good permeability. The authors argue that several parameters, such as the number of nucleation points and the scale factor, are the main factors in regulating the basic properties of the scaffold, such as porosity, specific area, and diameter of the supports, which are considered essential to achieve a good TE design. Ali et al. (2020) [30] looked at the permeability and wall shear stress of eight strut-based and triply periodic minimal surface (TPMS) structures at 80% porosity. These structures include the gyroid and the Schwarz P. Except for the gyroid model, which shows relative permeability, the strut-based scaffold has better permeability than the TPMS surface.

Tortuosity of flow paths and friction of the pore walls result in variations in the fluid medium motion, which then affect the usefulness of the scaffold in the TE. It is, therefore, important that in the development of 3D scaffolds, tortuosity and geometric curvature should be considered early on. It was further explained that tortuosity is an important parameter closely related to molecular diffusion, fluid permeation, and effective diffusivity transport behavior. This change is related to the design parameters as this will ultimately affect the petrophysical properties, such as the permeability of the designed TE scaffold. Many studies have been conducted on cell attachment, proliferation, and differentiation in tissue scaffolds and collective cell migration behavior, somewhat influenced by tortuous design parameters. Cell migration efficiency in scaffolds with tortuous architecture provides superior cell attachment compared to scaffolds having relatively straight microchannels [13]. The tortuous structure can better trap cultured cells and provide superior cell organization compared to straight microchannels [12,13]. The geometric curvature of the scaffold is thought to play an important role in the migration of tortuous microchannel cells in promoting bone tissue regeneration [12,42,43,44]. Modifying the surface curve where cells connect to the extracellular matrix will promote tissue development [20,44]. Mazalan et al. (2020) [12] established a tortuous microchannel device from polydimethylsiloxane (PDMS) to investigate collective cell migration under various geometric constraints, with a tortuosity index ranging from 1.57 to 2.30. The authors found that changing the radius of curvature and the tortuosity index resulted in a unique collective cell migration speed, thus further strengthening our argument that tortuosity is an important parameter in TE design. In addition, many researchers have considered general parameters in the design of tissue engineering scaffolds, including pore size, shape, porosity, surface area, trabecular thickness, and trabecular separation [20]. However, tortuosity parameters have yet to be explored more deeply in the designed scaffold.

Considering the factors described above, the solution given in this study is to consider a 3D open porous scaffold model based on meandering microchannels designed using computer software. Controlling the pore size, radius of curvature, and scaling size of scaffold morphology, such as porosity, strut thickness, tortuosity, and specific surface area, can be adjusted more effectively. The uniqueness of the proposed NSP model design would be to control the tortuosity of the scaffold by manipulating the pore size and curvature of the radius and, thus, the tortuosity of the scaffold. The pore size was chosen according to a design model known to promote cell proliferation and growth in the bone, which is 0.3–0.9 mm [45,46,47,48,49]. As previously explained, trabecular thickness Tb.Th and Tb.Sp is correlated with porosity (see Figure 4a); therefore, changes in this parameter will inherently regulate the existing pore size. It has been shown that by increasing the pore size and reducing the radius of curvature, the Tb.Th and Tb.Sp of the NSP model simultaneously increase to the desired level. The Tb.Th and Tb.Sp values of the proposed NSP model design are the same in the natural cancellous bone range (Tb.Th = 0.081–1.890 mm and Tb.Sp = 0.148–5.085 mm), similar to previous studies [50]. Mazalan et al. (2020) [12] proposed a PDMS-based microchannel with a tortuosity index of 1.57–2.30, giving the results of different migration velocities of the collective cells. These values are generally within the range obtained in our TE-designed constructs and healthy mammalian cancellous bone. In addition, the tortuosity of the NSP model is in the range of 1.31–2.40.

Nevertheless, the tortuous structure may not be the only predictive factor in favor of cell longevity. The specific surface area (BS/TV) factor is also important. Fyhri et al. (1999) [32] proposed that BS/TV affects the transport of metabolites between hard trabecular tissue and marrow. It is also mentioned that BS/TV plays an important role in improving cell migration and proliferation [51] because its larger surface area increases the percentage of cell migration and proliferation. Accordingly, our NSP scaffolds provide excellent specific surface area compared to cubic, octahedron pillar, and Schoen gyroid structures. The BS/TV value of the NSP scaffold is also in the same range as that of the cancellous bone of mammals, according to Adam et al. (2014) [33].

Computational fluid dynamic analysis is an important tool in bone tissue engineering, which requires precise microstructural configuration taking into account dynamic cellular parameters, such as cell proliferation growth, which will affect permeability and local shear stress [30,52]. In this study, the results of computational fluid dynamics simulations showed that the permeability values of the NSP scaffolds were similar to those of cancellous bone, as found in previously reported experimental investigations [5,24,32,33,35]. Schoen gyroid, pillar octahedron, cubic, cancellous, and NSP models produce different permeabilities for scaffolds with the same porosity and structure. This phenomenon occurs because the tortuosity parameter of a more tortuous porous structure will increase the fluid resistance when passing through the scaffold, so the pressure drop increases and decreases the permeability value. The relationship between tortuosity and permeability gives a value of R^2^ = 0.8 (see Figure 8). Thus, tortuosity is a very important parameter in designing a superior tissue scaffold because the parameter is the most influential in affecting scaffold permeability, as in previous investigations [18,19]. In addition, there is also a polynomial correlation between the tortuosity and permeability of cancellous bone in the same structure. Structures with the same porosity and surface area have different tortuosities in different orientations. This finding is similar to previous studies’ results, where the permeability value of cancellous bone is different in the longitudinal and transverse directions [53]. According to Rabiatul et al. (2021) [53], this phenomenon occurs due to the orientation of the trabecular strut effect, which causes different bone marrow permeability. We assume that tortuosity is also the dominant parameter in controlling flow’s effective transport and direction to hard and soft tissues, regulating bone remodeling and cartilage regeneration. Differences in cancellous bone tortuosity related to cell migration and bone regeneration must be investigated in future studies.

Another important parameter studied in scaffold design is the wall shear stress (WSS), which affects the performance of cells within the scaffold during bone growth. WSS occurs through the load-driven fluid flow of the relative motion between the scaffold and cells and tissues in the fluidic phase within the scaffold. Previous studies have shown that different WSS levels affect cellular differentiation and proliferation [54,55]. This study showed a significant correlation between WSS and tortuosity (see Figure 7b). This phenomenon occurs because tortuosity will result in an enlarged specific surface area and an increase in fluid frictional force, which have an impact on increasing the WSS value. In addition, WSS can be manipulated by changing the tortuosity without changing the scaffold’s porosity. This finding is considered important when designing bone scaffolds to enhance cell performance without affecting the porosity of the structure. The results of this study indicate that the WSS value of the NSP scaffold was obtained in the range of 0.8–3.0 Pa, which is still within the WSS range of 0.01–3.0 Pa, which will initiate the stem cell response as used in the literature [56,57].

The simulation showed that the CFD analysis obtained streamlines, velocity, and pressure distribution. The streamline showed that NSP was close to natural cancellous bone compared with cubic; the fluid can travel almost straight, meaning that the cells minimize interaction on the inner surface of the scaffold (cell migration–cubic scaffold). However, NSP scaffolds with more complicated internal curvature (tortuous) have paths that force the flow to interact with the scaffold wall, which makes them the least permeable of all scaffolds. The NSP scaffold is the most suitable for most tissue engineering applications. It has both low permeability and a travel path that leads the cells inside the flow to interact with the inner surface of the scaffold. In other words, it may increase the percentage of cell growth. Based on the present study, tortuosity can be manipulated by adjusting the radius of curvature surfaces of scaffold geometry.

However, the limitation of this study was that scaffolds were not subjected to in vivo testing for mechanical properties, degradation, fatigue and vascularity due to their limited size and material. Mechanical properties are essential in load-bearing conditions for providing structural support. Therefore, scaffolds must possess enough mechanical properties to match those of the cancellous bone and should have enough strength when implanted and when the material degrades over time while remaining in contact with the bone marrow [58,59]. This limitation can be overcome by 3D printing the scaffold using biopolymer in the size of a specimen testing standard and the material of an in vivo standard specimen to mimic the native anatomy.

## 5. Conclusions

This study demonstrates that modifying the pore size and radius of curvature of the TE scaffold improves the porosity and permeability of the structure. Therefore, modifying the scaffold’s tortuosity can achieve the desired bone tissue engineering structure. Thus, it is possible to achieve improvements in supporting cell migration and tissue regeneration. With the NSP model’s tortuosity ranging from 1.31 to 2.40, the permeability will be between 2.0 × 10^−11^ m^2^ and 4.0 × 10^−10^ m^2^. The resultant permeability value suggests that it is comparable to cancellous bone permeability. Therefore, it is suggested that the NSP model’s tortuosity be considered when designing new scaffolds. In this study, an attempt was made to make a new scaffold similar in structure and function to natural cancellous bone.

## Figures and Tables

**Figure 1 biomedicines-11-00427-f001:**
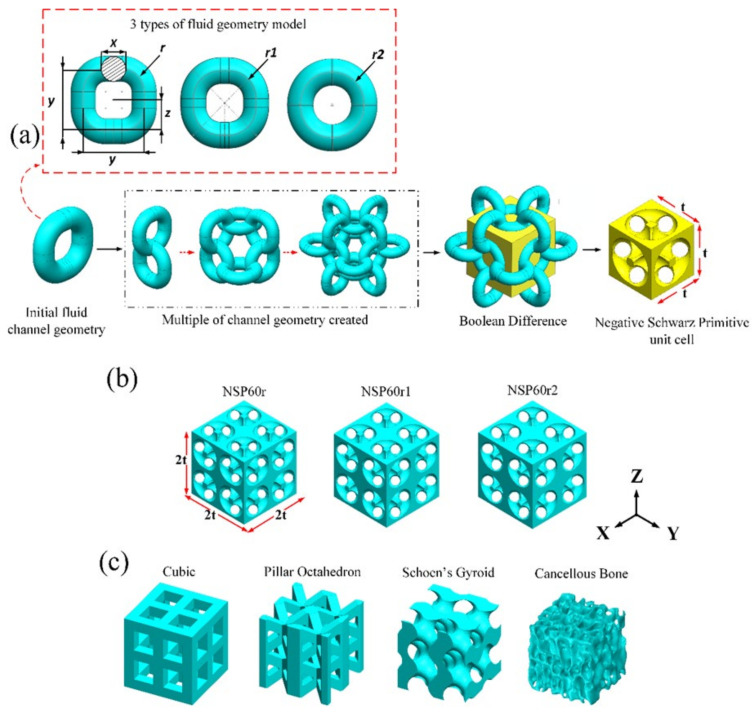
Schematic illustration of the scaffold design stage: (**a**) process schematic diagram showing the principles of scaffold design based on the concept of meandering pore channels; (**b**) NSP scaffolds with 60% porosity and different tortuosities can be controlled by adjusting the fluid pore size and radius of curvature; (**c**) cubic, PO, SG and natural cancellous bone models with 60% porosity.

**Figure 2 biomedicines-11-00427-f002:**
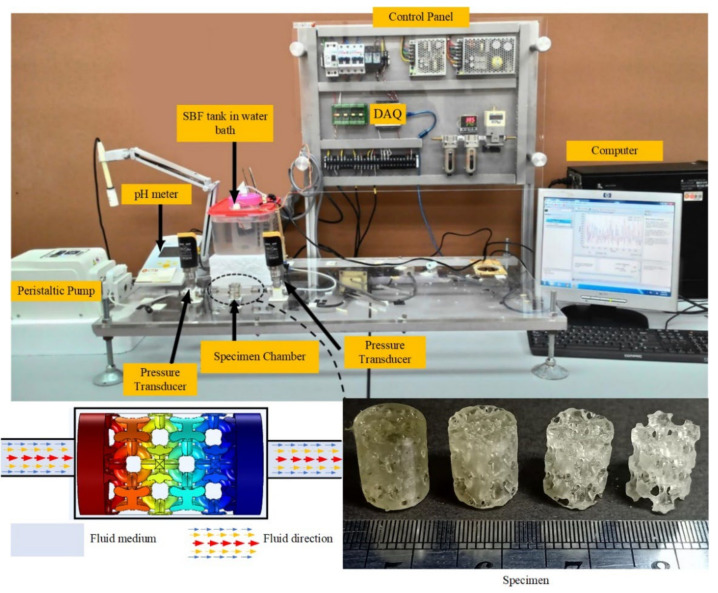
Flow simulation apparatus for testing the permeability of porous structure.

**Figure 3 biomedicines-11-00427-f003:**
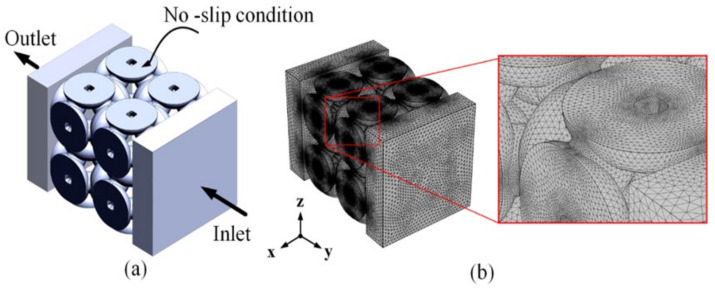
The steps used to characterize the fluid dynamics of the scaffold: (**a**) The simulation’s boundary condition; (**b**) a suitable convergent meshing model can be obtained.

**Figure 5 biomedicines-11-00427-f005:**
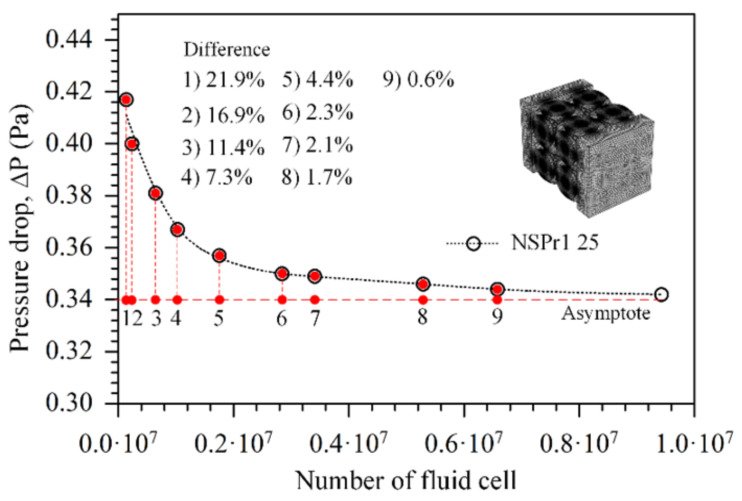
Mesh convergence study on one model porous scaffold NSPr1 with porosity of 25%.

**Figure 6 biomedicines-11-00427-f006:**
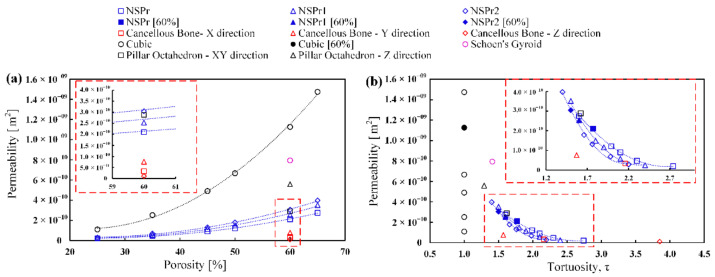
Relationship between permeability and: (**a**) porosity scaffold; (**b**) tortuosity scaffold.

**Figure 7 biomedicines-11-00427-f007:**
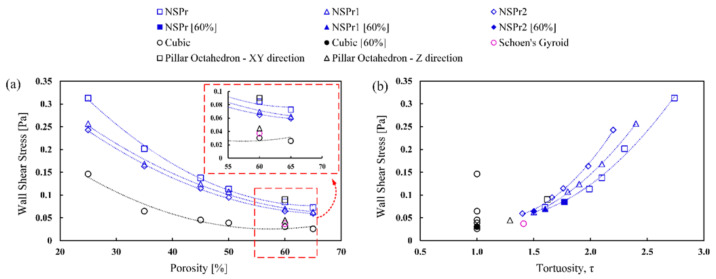
Relationship between WSS and: (**a**) porosity scaffold; (**b**) tortuosity scaffold.

**Figure 8 biomedicines-11-00427-f008:**
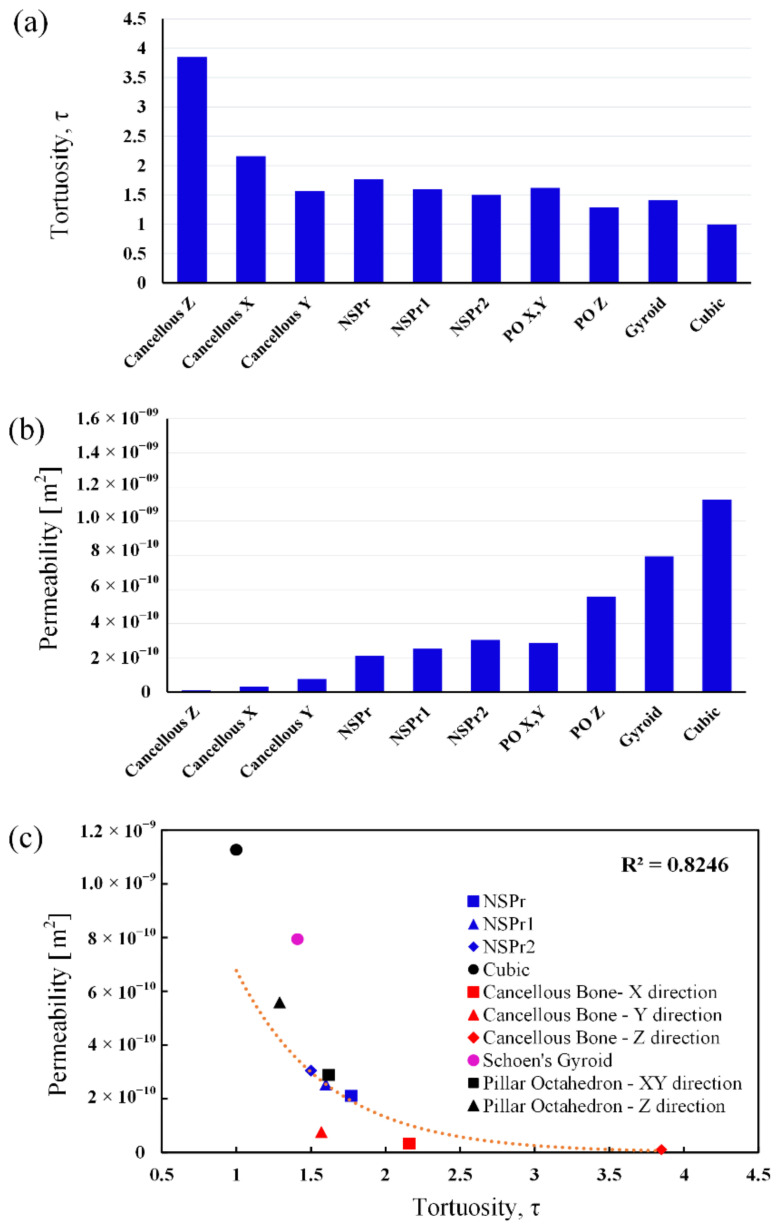
Characterization of various scaffold structures: (**a**) tortuosity of different structures with the same porosity (60%); (**b**) permeability of different structures with the same porosity (60%); (**c**) the relationship between permeability and tortuosity of different structures with the same porosity (60%).

**Figure 9 biomedicines-11-00427-f009:**
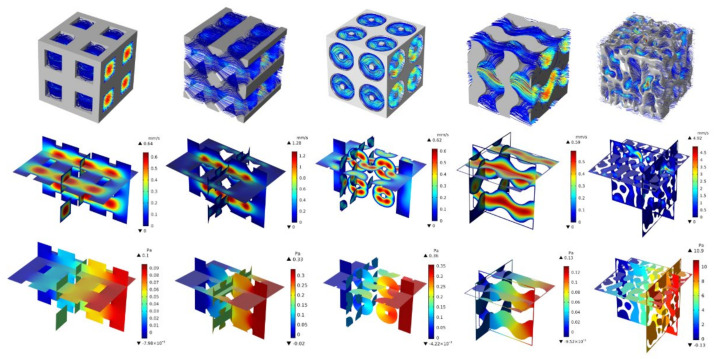
Velocity streamlines, velocity contour, and pressure of porous structure with 60% porosity.

**Figure 10 biomedicines-11-00427-f010:**
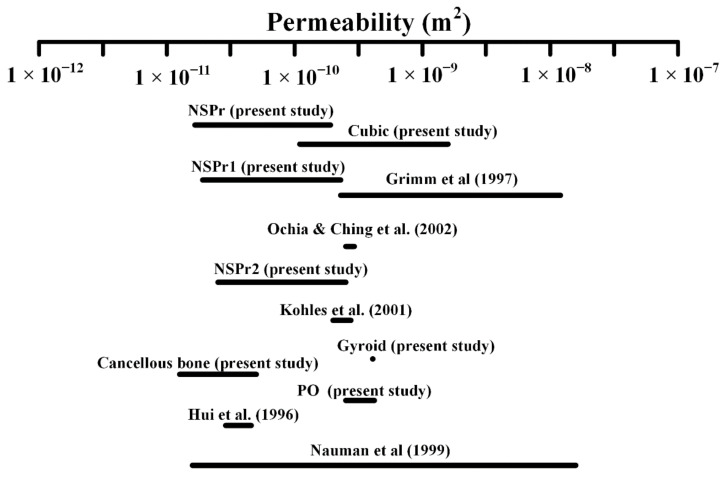
Permeability values of cancellous bone were discovered in the literature and evaluated in this study [24,35,36,37,38].

**Table 1 biomedicines-11-00427-t001:** Dimension parameter features of the NSP model CAD design (see also Figure 1a).

Dimensional Parametric Study	Value (mm)		
Model	NSPr	NSPr1	NSPr2
r	0.4	0.45	0.49
Constant, c	0.745	0.745	0.745
y	2c−r	2c−r	2c−r
z	$c-\frac{r}{2}$	$c-\frac{r}{2}$	$c-\frac{r}{2}$
t	2.1	2.1	2.1
X(Φ: 25%)	0.450	0.421	0.41
X(Φ: 35%)	0.540	0.515	0.51
X(Φ: 45%)	0.620	0.603	0.600
X(Φ: 50%)	0.660	0.646	0.645
X(Φ: 60%)	0.740	0.731	0.735
X(Φ: 65%)	0.780	0.775	0.778

NSPr = Negative Schwarz Primitive Scaffold model with radius curvature 0.4 mm; NSPr1 = Negative Schwarz Primitive Scaffold model with radius curvature 0.45 mm; NSPr2 = Negative Schwarz Primitive Scaffold model with radius curvature 0.49 mm.

**Table 2 biomedicines-11-00427-t002:** Permeability value of the NSPr model, as determined by both experimental and simulation results.

Porosity [%]	Permeability			
	Experimental			Simulation $k\times 10^{-10}\left(m^{2}\right)$
	Mean $k\times 10^{-10}\left(m^{2}\right)$	Standard Deviation $\sigma \times 10^{-10}\left(m^{2}\right)$	*p*-Value	
25	0.152	0.014	0.00287	0.148
45	1.014	0.694	0.00537	1.052
60	2.655	0.583	0.01571	2.537
65	3.143	0.976	0.03071	3.315

## Data Availability

The data presented in this study are available on request from the corresponding author.

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
