# Peer review of "The Effect of Tortuosity on Permeability of Porous Scaffold"

_biomedicines, 2023, doi:10.3390/biomedicines11020427_

Round 1

Reviewer 1 Report

The current manuscript provides an interesting account of the effect of tortuosity on permeability of porous scaffold and provides important insights into the morphological properties of scaffolds for various biomedical applications.

The use of a flow simulation apparatus for testing the permeability of porous structure is very unique and I see this as a trend setter.

I suggest following revisions for the manuscript:

1. The physical morphological features using SEM should have been provided. The authors have tried to do this but the results are not provided.

2. I am surprised that the image processing of the morphological fetaures are not conducted using imageJ or other programs.

3. The physical porosity analysis using BET isotherm should have been provided.

Author Response

Response to Reviewer 1 Comments

Point 1: The physical morphological features using SEM should have been provided. The authors have tried to do this but the results are not provided.

Response 1: Many acknowledge the reviewers' excellent evaluation of the current manuscript. In this study, we did not access physical morphology. However, we focus on structure morphology.

Point 2: I am surprised that the image processing of the morphological features are not conducted using imageJ or other programs.

Response 2: Many acknowledge the reviewer's excellent evaluation of this present manuscript. We accessed the morphology using image processing, which was conducted using the Fiji (Image J, NIH) and Bone J plugins. The morphology indices included trabecular thickness (Tb.Th) and separation (Tb.Sp), porosity, and surface area. The tortuosity of cancellous bone and scaffold was determined using the MATLAB Software integrated Taufactor plugin. The method of the present study has been revised as follows:

Line 131-141

The scaffold morphology, including porosity and surface area, was conducted using CAD software features, and the cancellous bone was modeled using mimics software features. 3D CAD models were exported to stereolithography (STL) format and imported into the slice software program Chitubox (CBD-Tech, Guangdong, China). The CAD model was sliced using a resolution of 17.20 µm. The remaining 244 slices and 500 x 500 pixel images were analyzed using Fiji (Image J, NIH). The trabecular thickness (Tb.Th) and separation (Tb.Sp) were calculated using the Fiji and BoneJ plugins. The image data set was then exported to MATLAB (MathWorks Corp., Natick, MA, USA) software to calculate the diffusion tortuosity. The open solver plugin Taufactor calculates the tortuosity factor based on the finite difference method (FDM) and directly uses image voxels as discretization meshes for simulation [22,23].

Point 3: The physical porosity analysis using BET isotherm should have been provided.

Response 3: Thank you for your comment and guidance. In this study, we conduct porosity and surface area measurements using image processing using Image J and CAD features. We know BET Isotherm can analyze the surface area and porosity.

Reviewer 2 Report

The scientific paper "The Effect of Tortuosity on Permeability of Porous Scaffold" aimed to develop a porous scaffold according to the morphological indices of cancellous bone (porosity, specific surface area, thickness and tortuosity).

It can be considered that:

1)      In the introduction, on lines 60 and 61, the author describes that "Many studies have shown...." but references only one study, number 11. Please adjust by including more references.

2)      In the third paragraph of the introduction, lines 69-76, there is no reference. Please include.

3)      Was the study submitted to an ethics committee for using cancellous bone samples from the bovine lateral condyle area? If yes, please include committee and approval number.

4)      Below table 1, add a caption with the meaning of the abbreviations: NSPr, NSPr1 and NSPr2

5)      In the text of the entire manuscript, for each citation of a product used, describe the manufacturer, city and country. Several citations in the text are incomplete. I can cite an example: line 134: (FL0001-EMA Electronic Ltd.)

6)      Authors should pay attention to the fact that each abbreviation, in its first citation, must include its full meaning. I can cite an example where there are flaws in the manuscript, as in line 136, with the abbreviation PTFE (missing polytetrafluoroethylene)

7)      I suggest that figure 2 be placed as a "supplementary file"

8)      In the discussion (throughout your text), adjust citations that include the authors' names. Include year of publication. As an example, adjust on line 314 and 315:

de: H. Chen et al., Fantini et al., and Gomez et al. [34–36]

to: Chen et al. (2021) [34], Fantini et al. (2018) [35] and Gomez et al. (2016) [36].....

Author Response

Response to Reviewer 2 Comments

Point 1: In the introduction, on lines 60 and 61, the author describes that "Many studies have shown...." but references only one study, number 11. Please adjust by including more references..

Response 1:

Thank you very much for your suggestion. As the reviewer suggested, the manuscript was checked and suggested corrections were made. The second paragraph of the introduction has been added, with the reference as follows:

Line 58-61

Many studies have shown that tortuous models during tissue growth promote better cell anchoring and tissue repair [11–14]. Furthermore, it has been shown that scaffolds with tortuous architecture provide better cell attachment than scaffolds with relatively straight microchannels [13].

References:

  1. Innocentini, M.D.M.; Faleiros, R.K.; Pisani, R.; Thijs, I.; Luyten, J.; Mullens, S. Permeability of Porous Gelcast Scaffolds for Bone Tissue Engineering. J. Porous Mater. 2010, 17, 615–627, doi:10.1007/s10934-009-9331-2.
  2. Mazalan, M. Bin; Ramlan, M.A. Bin; Shin, J.H.; Ohashi, T. Effect of Geometric Curvature on Collective Cell Migration in Tortuous Microchannel Devices. Micromachines 2020, 11, 1–17, doi:10.3390/MI11070659.
  3. Ali, D. Effect of Scaffold Architecture on Cell Seeding Efficiency: A Discrete Phase Model CFD Analysis. Comput. Biol. Med. 2019, 109, 62–69, doi:10.1016/j.compbiomed.2019.04.025.
  4. Zhang, L.; Song, B.; Yang, L.; Shi, Y. Tailored Mechanical Response and Mass Transport Characteristic of Selective Laser Melted Porous Metallic Biomaterials for Bone Scaffolds. Acta Biomater. 2020, 112, 298–315, doi:https://doi.org/10.1016/j.actbio.2020.05.038.

Point 2: In the third paragraph of the introduction, lines 69-76, there is no reference. Please include.

Response 2:

Thank you very much for your suggestion. As the reviewer suggested, the manuscript was checked and suggested corrections were made. The third paragraph of the introduction has been added, with the reference as follows:

Line 67 -74

While the role of scaffold permeability has been considered necessary in bone replacement design, the tortuosity's importance has yet to be further elaborated [17,18]. It has been shown that permeability is positively related to porosity or pore size but negatively related to an increase in the surface area of the scaffold [19–21]. In addition, based on the fact that although tortuosity is directly related to permeability, controlling one of the parameters using a simple approach has yet to be carried out. So far, the direct relationship between tortuosity and permeability of scaffold structures has yet to be studied before and therefore requires further investigation.

References:

  1. Guerreiro, R.; Pires, T.; Guedes, J.M.; Fernandes, P.R.; Castro, A.P.G. On the Tortuosity of TPMS Scaffolds for Tissue Engineering. Symmetry (Basel). 2020, doi:10.3390/SYM12040596.
  2. Fiume, E.; Schiavi, A.; Orlygsson, G.; Bignardi, C.; Verné, E.; Baino, F. Comprehensive Assessment of Bioactive Glass and Glass-Ceramic Scaffold Permeability: Experimental Measurements by Pressure Wave Drop, Modelling and Computed Tomography-Based Analysis. Acta Biomater. 2021, doi:10.1016/j.actbio.2020.10.027.
  3. Schiavi, A.; Fiume, E.; Orlygsson, G.; Schwentenwein, M.; Verné, E.; Baino, F. High-Reliability Data Processing and Calculation of Microstructural Parameters in Hydroxyapatite Scaffolds Produced by Vat Photopolymerization. J. Eur. Ceram. Soc. 2022, 42, 6206–6212, doi:https://doi.org/10.1016/j.jeurceramsoc.2022.06.022.
  4. Syahrom, A.; Abdul Kadir, M.R.; Abdullah, J.; Öchsner, A. Permeability Studies of Artificial and Natural Cancellous Bone Structures. Med. Eng. Phys. 2013, 35, 792–799, doi:10.1016/j.medengphy.2012.08.011.
  5. Syahrom, A.; Abdul Kadir, M.R.; Harun, M.N.; Öchsner, A. Permeability Study of Cancellous Bone and Its Idealised Structures. Med. Eng. Phys. 2015, 37, doi:10.1016/j.medengphy.2014.11.001.

Point 3: Was the study submitted to an ethics committee for using cancellous bone samples from the bovine lateral condyle area? If yes, please include committee and approval number.

Response 3:

Thank you very much for your suggestion. However, we do not apply ethical approval to the ethics committee in this study. For your information, we used a bovine's cancellous bone sample, which was gathered from a local slaughterhouse. Then, it is optional for us to apply for ethical approval from the ethics committee.

Line 83-109

A bovine lateral femoral condyle was freshly gathered from a local slaughterhouse for a cancellous bone sample. The bovine lateral femoral condyle was cut into rectangular specimens using a low-speed (150 rpm) diamond saw (Behringer GmbH, Type SLB 230 DG HA, Kirchardt) under constant lubricant irrigation to minimize structural breakage and heat generation. Saline water was used as a lubricant to ensure that the temperature did not exceed 45 °C to protect the cancellous bone sample from heat damage. The sample temperature was constantly observed based on the blade and coring bit using infrared thermometer (Fluke 62 Mini Infrared Thermometer). In addition, the cutting process was stopped at several stages to ensure that the temperature did not exceed a critical level. At 150–250 rpm, a 1.5 mini-thick diamond tip coring bit was used to drill the trabecular sample into a cylindrical shape with a total length 12 mm and a diameter of 10 mm. The sample then cleaned by soaking the cancellous bone sample in chemical detergent (Pumicized, Gent-1-Kleen) using an ultrasonic cleaner (Cest Ultrasonic, model P1100SR, USA) for 3 hours. The procedure continues with further cleansing to remove any excess bone marrow, water and fat using water jets and air jets until any excess bone marrow cannot be detected upon visual inspection. The cancellous bone sample was then stored in an airtight plastic bag to minimize the thermal cycling and put in the freezer at a temperature of -20 °C. The sample was then scanned via high-resolution micro-computed tomography (μCT) images (Skyscan 1172™, Bruker micro-CT, Kontich, Belgium) with a voltage source of 100 kV, a current of 100 µA and a resolution of 17.20 µm. The tomographic reconstruction of these images using Materialize Mimics® gives a three-dimension (3D) image volume dataset of 712 layers of two-dimension (2D) images of the cancellous bone structure. This dataset is 12 mm in height and 10 mm in diameter. The software extracted the desired region (ROI) with 60% porosity from the sample. Fi-nally, three different scaffold structures (cubic, octahedron pillar (PO), and Schoen's Gyroid (SG)) and cancellous bone structures with a porosity (Φ) of 60% were developed and compared.

Point 4: Below table 1, add a caption with the meaning of the abbreviations: NSPr, NSPr1 and NSPr2

Response 4:

Many thanks to the reviewer for your excellent assessment of the current manuscript. The meanings of the abbreviations below Table 1 have been updated to include NSPr, NSPr1, and NSPr2 as follows:

Line 129

Table 1. Dimension parameter features of the NSP model CAD design

NSPr  = Negative Schwarz Primitive Scaffold model with radius curvature 0.4 mm

NSPr1 = Negative Schwarz Primitive Scaffold model with radius curvature 0.45 mm

NSPr2 = Negative Schwarz Primitive Scaffold model with radius curvature 0.49 mm

Point 5: In the text of the entire manuscript, for each citation of a product used, describe the manufacturer, city and country. Several citations in the text are incomplete. I can cite an example: line 134: (FL0001-EMA Electronic Ltd.)

Response 5:

Many thanks go to the reviewer for thoroughly assessing the current manuscript. We have completed the revision; please refer to lines 84–87; 88–90; 94–96; 100–102; 102–106; 110–112; 133–134; 137–139; 151–152; 158–160; 160–161, 166–167; 180–182 of the revised manuscript as follows:

Line 84-87

The bovine lateral femoral condyle was cut into rectangular specimens using a low-speed (150 rpm) diamond saw (Behringer GmbH, Type SLB 230 DG HA, Kirchardt) under constant lubricant irrigation to minimize structural breakage and heat generation.

Line 88-90

The sample temperature was constantly observed based on the blade and coring bit using Fluke 62 Mini Infrared Thermometer (FLUKE Europe, Eindhoven, Netherlands).

Line 94-96

The sample then cleaned by soaking the cancellous bone sample in chemical detergent (Pumicized, Gent-1-Kleen,USA) using an ultrasonic cleaner (Cest Ultrasonic, model P1100SR, USA) for 3 hours.

100-102

The sample was then scanned via high-resolution micro-computed tomography (μCT) images (Skyscan 1172, Bruker micro-CT, Kontich, Belgium) with a voltage source of 100 kV, a current of 100 µA and a resolution of 17.20 µm.

102-106

The tomographic reconstruction of these images using Materialize Mimics® software (Materialize, Wilfried, Leuven, Belgium) gives a three-dimension (3D) image volume dataset of 712 layers of two-dimension (2D) images of the cancellous bone structure. This dataset is 12 mm in height and 10 mm in diameter.

Line 110-112

In this study, the scaffold is an open porous model with tortuous microchannels made with SolidWorks (Dassault Systèmes SolidWorks Corp., Waltham, MA, USA) software.

Line 133-134

3D CAD models were exported to stereolithography (STL) format and imported into slice software programs Chitubox (CBD-Tech, Guangdong, China).

Line 137-139

The image data set was then exported to MATLAB (MathWorks Corp., Natick, MA, USA) software to calculate the diffusion tortuosity.

Line 151-152

The mass flow rate is measured directly from the tube using an intelligent flow sensor FL0001 (EMA Electronic Ltd, New Taipei City, Taiwan).

Line 158-160

The differential pressure is measured using a pressure transducer pressure sensor (EMA Electronic Ltd, New Taipei City, Taiwan) with a maximum pressure of 1 bar.

Line 160-161

The USB-6009 instrument data acquisition system (DAQ) (National Instruments company, Austin, TX, USA) was used to read, collect and store data during the test.

Line 166-167

Computational simulations were performed using COMSOL Multiphysics® Soft-ware (COMSOL, Inc., Burlington, MA, USA).

Line 180-182

This simulation was carried out on a Personal Computer (Dell Precision 3630 Tower Workstation, TX, USA) with an Intel I7-8700 processor and 80 GB of RAM.

Point 6: Authors should pay attention to the fact that each abbreviation, in its first citation, must include its full meaning. I can cite an example where there are flaws in the manuscript, as in line 136, with the abbreviation PTFE (missing polytetrafluoroethylene)

Response 6:

Thanks to the reviewer for their comments on the present manuscript. The revisions were done; please refer to the manuscript as follows:

Line 102-105

The tomographic reconstruction of these images using Materialize Mimics® software (Materialize, Wilfried, Leuven, Belgium) gives a three-dimension (3D) image volume dataset of 712 layers of two-dimension (2D) images of the cancellous bone structure.

Line 129

Table 1. Dimension parameter features of the NSP model CAD design (see also Figure 1a).

NSPr  = Negative Schwarz Primitive Scaffold model with radius curvature 0.4 mm

NSPr1 = Negative Schwarz Primitive Scaffold model with radius curvature 0.45 mm

NSPr2 = Negative Schwarz Primitive Scaffold model with radius curvature 0.49 mm

Line 154-156

The peripheral surface of the specimen is wrapped with Polytetrafluoroethylene (PTFE) tape to prevent fluid flow along the outer surface of the specimen from entering.

Line 172-174

This simulation used Simulated Body Fluid (SBF) liquid with a viscosity of 1 mPa.s and a density of 1 g/cm3 at body temperature (37oC) [27].

Line 174-176

The boundary between the fluid and solid is characterized non-slip boundary during the Computational Fluid Dynamic (CFD) study where the fluid velocity at the boundary is equal to the velocity of the solid [28].

Line 375-377

Mazalan et al. (2020) [12] established a tortuous microchannel device from Polydimethylsiloxane (PDMS) to investigate collective cell migration under various geometric constraints, with a tortuosity index ranging from 1.57 to 2.30.

Point 7: I suggest that figure 2 be placed as a "supplementary file"

Response 7:

Thank you for the comment and suggestion. However, in Figure 2, we are thinking of making it easier for the reader to refer to the supplementary file.

Point 8: In the discussion (throughout your text), adjust citations that include the authors' names. Include year of publication. As an example, adjust on line 314 and 315: de: H. Chen et al., Fantini et al., and Gomez et al. [34–36] to: Chen et al. (2021) [34], Fantini et al. (2018) [35] and Gomez et al. (2016) [36]....

Response 8:

Thanks to the reviewer for their comments on the present manuscript. The authors' citations have been adjusted to include the author's name and year of publication as follows:

Line 347-351

Many researchers have previously claimed to have developed such a scaffold. For example, H. Chen et al. (2021) [40], Fantini et al. (2018) [41], and Gomez et al. (2016) [42] have built irregular scaffolds using Voronoi tessellation and shown that successful designs have good permeability.

Line 354-356

  1. Ali et al. (2020) [30] looked at the permeability and wall shear stress of eight strut-based and Triply Periodic Minimal Surface (TPMS) structures at 80% porosity.

Line 375-377

Mazalan et al. (2020) [12] established a tortuous microchannel device from Polydimethylsiloxane (PDMS) to investigate collective cell migration under various geometric constraints, with a tortuosity index ranging from 1.57 to 2.30.

Line 398-403

Mazalan et al. (2020) [12] proposed a PDMS-based microchannel with a tortuosity index of 1.57 – 2.30, giving the results of different migration velocities of the collective cells.

Line 405-407

D.P. Fyhri et al. (1999) [32] proposed that BS/TV affects the transport of metabolites between hard trabecular tissue and marrow.

Line 411-412

Also, the BS/TV value of the NSP scaffold is in the same range as that of the cancellous bone of mammals, according to Adam et al. (2014) [33].

Line 431-433

According to Rabiatul et al. (2021) [57], this phenomenon occurs due to the orientation of the trabecular strut effect, which causes different bone marrow permeability.

Reviewer 3 Report

This paper is appropriate for the Journal, well written and of high interest in the biomaterials arena. Therefore, I strongly recommend publication. I have only a couple of minor suggestions:

1. these recent studies on the assessment of scaffold permeability deserve to be cited:

 Comprehensive assessment of bioactive glass and glass-ceramic scaffold permeability: experimental measurements by pressure wave drop, modelling and computed tomography-based analysis. Acta Biomater. 2021;119:405-418

High-reliability data processing and calculation of microstructural parameters in hydroxyapatite scaffolds produced by vat photopolymerization. J. Eur. Ceram. Soc. 2022;42:6206-6212

2. Experimental results (fluid-flow experiments) for permeability should be reported as mean +/- SD in a table. Statistical analysis (at least p-values) should be considered as well.

Author Response

Response to Reviewer 3 Comments

Point 1: These recent studies on the assessment of scaffold permeability deserve to be cited:

Comprehensive assessment of bioactive glass and glass-ceramic scaffold permeability: experimental measurements by pressure wave drop, modelling and computed tomography-based analysis. Acta Biomater. 2021;119:405-418

High-reliability data processing and calculation of microstructural parameters in hydroxyapatite scaffolds produced by vat photopolymerization. J. Eur. Ceram. Soc. 2022;42:6206-6212

Response 1:

Thank you very much for your suggestion. The reference was updated, and all suggestion papers were cited. Please refer below and to the manuscript:

Line 67-74

While the role of scaffold permeability has been considered necessary in bone replacement design, the tortuosity's importance has yet to be further elaborated [17,18]. It has been shown that permeability is positively related to porosity or pore size but negatively related to an increase in the surface area of the scaffold [19–21]. In addition, based on the fact that although tortuosity is directly related to permeability, controlling one of the parameters using a simple approach has yet to be carried out. So far, the direct relationship between tortuosity and permeability of scaffold structures has yet to be studied before and therefore requires further investigation.

Line 424-426

Thus, tortuosity is a very important parameter in designing a superior tissue scaffold because the parameter most influent in affecting scaffold permeability same as previous investigation [18][19].

References:

  1. Fiume, E.; Schiavi, A.; Orlygsson, G.; Bignardi, C.; Verné, E.; Baino, F. Comprehensive Assessment of Bioactive Glass and Glass-Ceramic Scaffold Permeability: Experimental Measurements by Pressure Wave Drop, Modelling and Computed Tomography-Based Analysis. Acta Biomater. 2021, doi:10.1016/j.actbio.2020.10.027.
  2. Schiavi, A.; Fiume, E.; Orlygsson, G.; Schwentenwein, M.; Verné, E.; Baino, F. High-Reliability Data Processing and Calculation of Microstructural Parameters in Hydroxyapatite Scaffolds Produced by Vat Photopolymerization. J. Eur. Ceram. Soc. 2022, 42, 6206–6212, doi:https://doi.org/10.1016/j.jeurceramsoc.2022.06.022.

Point 2: Experimental results (fluid-flow experiments) for permeability should be reported as mean +/- SD in a table. Statistical analysis (at least p-values) should be considered as well.

Response 2:

Many thanks go to the reviewer for thoroughly assessing the current manuscript. As the reviewer suggested, the manuscript was checked and suggested corrections were made. The permeability value was determined from experimental and simulation results, as tabulated in Table 2.

 Line 196-202

2.5. Statistical Analysis

The permeability of scaffolds was measured three times on four samples with porosities of 25%, 45%, 60%, and 65% to accurately assess the reproducibility and repeatability of the results and compare them with simulation results. The experimental results were expressed in the average value (k) and standard deviation (σ). The statistical analysis was conducted using MINITAB software (Minitab Inc., State College, PA, USA).

Line 245-248

The permeability value of the NSPr model, as determined by both experiment and simulation as tabulated in Table 2. The permeability value determined through computer simulation is comparable to or near to the value obtained from experimental measurements. The error in the permeability value between the simulation and experiment is below 5%.

Line 257-259

Table 2. Permeability value of the NSPr model, as determined by both experimental and simulation results

Round 2

Reviewer 2 Report

No comments